# A randomised controlled trial of fluoxetine versus naltrexone in compulsive sexual behaviour disorder: presentation of the study protocol

Josephine Savard  ,[1,2] Katarina Görts Öberg,[2,3] Cecilia Dhejne,[2,3] Jussi Jokinen[1,4]

¹Department of Clinical Sciences/Psychiatry, Umea University, Umea, Sweden
²ANOVA clinic, Karolinska University Hospital, Stockholm, Sweden
³Department of Medicine, Karolinska Institutet, Stockholm, Sweden
⁴Centre for Psychiatry Research, Department of Clinical Neuroscience, Karolinska Institutet, Stockholm, Sweden

**Correspondence to**
Josephine Savard;
josephine.savard@umu.se

## ABSTRACT

**Background** Compulsive sexual behaviour disorder is a new disorder in the International Classification of Diseases (ICD-11), and is associated with negative consequences in different areas of life. Evidence for pharmacological treatment of compulsive sexual behaviour disorder is weak and treatment options are limited. This proposed study will be the largest and the first randomised controlled trial comparing the efficacy and tolerability of two active drugs in compulsive sexual behaviour disorder.

**Methods and analysis** Eighty adult participants with compulsive sexual behaviour disorder according to ICD-11 will be randomised to receive either naltrexone 25–50 mg or fluoxetine 20–40 mg for 8 weeks, followed by 6 weeks without treatment. The study will be conducted in a subspecialised outpatient sexual medicine unit at Karolinska University Hospital, Stockholm, Sweden. The study is financed by grants and entirely independent of the manufacturers.

Exclusion criteria include severe psychiatric or psychical illness, changes to concurrent medication and non-compatible factors contraindicating the use of either drug. The primary outcome measure is the Hypersexual Disorder: Current Assessment Scale (HD: CAS), and tolerability will be assessed by the Udvalg for Kliniske Undersogelser side effect rating scale (UKU), drug accountability, adherence to treatment and drop-out rate. Participants will complete questionnaires at regular intervals, with the main endpoint for efficacy after 8 weeks (end of treatment) and after 14 weeks (follow-up). Blood chemistry will be repeatedly collected as a safety precaution and for research purposes. The results will be analysed using an appropriate analysis of variance model or a mixed model, depending on the distribution of HD: CAS and the extent of missing data.

**Ethics and dissemination** The Swedish Ethical Review Authority and the Swedish Medical Products Agency have approved the study on 27 May 2020 and 4 June 2020, respectively (ref. no. 2020-02069 and ref. no. 5.1-2020-48282). Findings will be published in peer-reviewed journals and presented at relevant conferences.

**Trial registration number** 2019-004255-36

### STRENGTHS AND LIMITATIONS OF THIS STUDY

⇒ This is the first randomised controlled trial comparing the efficacy of two pharmacological agents in compulsive sexual behaviour disorder—naltrexone and fluoxetine.
⇒ The study enables assessment of clinical, psychosocial and biological predictors of treatment response.
⇒ The lack of placebo-control is a limitation.
⇒ The study protocol duration of 14 weeks may be seen as limitation of the study design.

impulse control disorder section of the 11th edition of the International Classification of Diseases (ICD-11).[1] It entails a persistent pattern of failure to control intense repetitive sexual impulses or urges, resulting in sexual behaviours with negative consequences affecting different areas of life. Adverse consequences associated with CSBD include distress, unwanted pregnancies, sexually transmitted diseases, relationship problems, financial expenses, occupational impairment and risk of crime. Considering these potential severe consequences and the suggested prevalence of 3%–6% of the population,[2] there is an undisputable, urgent need for effective treatments.

Existing drug studies of CSBD have mainly been case reports and small open-label studies on selective serotonin reuptake inhibitors (SSRIs),[3 4] antiandrogens[4] or the opioid antagonist naltrexone.[5–8] Only one randomised controlled trial has examined the effect of an SSRI (citalopram 20–60 mg/day for 12 weeks) in 28 gay and bisexual men with non-paraphilic compulsive sexual behaviour.[9] There was no significant difference in the main outcome measure Yale-Brown Obsessive Compulsive Scale-Compulsive Sexual Behavior (Y-BOCS-CSB) score between those assigned citalopram and those assigned placebo.

## INTRODUCTION

Compulsive sexual behaviour disorder (CSBD) is a newly defined diagnosis in the

CSBD and paraphilic disorders are regarded as separate conditions, however, a high level of sexual preoccupation is common in both.[10–13] To aid physicians in clinical practice, the World Federation of Societies of Biological Psychiatry (WFSBP) has provided treatment guidelines for paraphilic disorders.[14 15] Psychotherapy is proposed as a first step for low-risk individuals. If results are unsatisfactory, the next step includes SSRIs.

The subsequent levels in the guidelines are for individuals with moderate-high risk for sexual violence and include testosterone-lowering agents such as cyproterone acetate and gonadotropin-releasing hormone agonists. These agents decrease the frequency and intensity of sexual desire and arousal, however use is associated with high rates of adverse effects. Hormonal treatment is potentially unsuitable for help-seeking individuals with conventional CSBD. In order to investigate appropriate pharmacological alternatives, the proposed 14-week randomised controlled trial will compare the efficacy of two active drugs in 80 adults with CSBD. Additional aims are to examine tolerability and what clinical characteristics and biomarkers could be predictors of response.

Based on the WFSBP treatment guidelines and available research,[9] the SSRI *fluoxetine* will be defined as standard treatment. Alternative treatment will consist of *naltrexone*, an opioid antagonist which prevents reinforcing effects in the mesolimbic reward centre and is used in the treatment of alcohol use disorder.[16] Naltrexone was chosen due to similarities between CSBD and other urge-driven disorders, and promising results in CSBD case reports, and in our pilot study of 20 men conducted during 2018–2019.[17] In the latter, all participants completed the 8-week study protocol, and no serious adverse events occurred. Although adverse reactions were common, all were considered mild to moderate, and most were transient during the first days-week. Hence, we were able to determine that naltrexone was a well-tolerated, feasible treatment option and gave indications of symptom relief. However, as the study format prohibited conclusion of efficacy, this study aims to further investigate naltrexone in the treatment of CSBD.

### Primary objective
To compare the effect of naltrexone versus fluoxetine in CSBD.

### Secondary objectives
To investigate if clinical, psychosocial or biological factors can predict treatment response.

To compare if there are any differences in drop-out rate and adherence between the two treatment groups.

To assess side effects and investigate any connection between drop-out rate, reports of side effects and/or efficacy.

To compare if there is a difference in the participants' wish to resume treatment.

To investigate biological markers and clinical parameters in participants with CSBD.

To investigate correlation between CSBD and impulsiveness, experience of violence and suicidality.

## METHODS AND ANALYSIS
### Study setting and study design
The study will be conducted at ANOVA, a subspecialised sexual medicine outpatient clinic at Karolinska University Hospital in Stockholm, Sweden. Eligible participants will be randomised in blocks with a 1:1 allocation to receive either fluoxetine or naltrexone in a superiority parallel group design for 8 weeks followed by a 6-week follow-up phase. Blinding will not be applicable due to the different appearances of the drugs and dose augmentation in different time intervals due to different pharmacological properties.

### Protocol version
Version identifier: 2.0. Number of protocol amendments: 1; 25 June 2020; main changes included adding blood samples and assessments for safety reasons as requested by the Swedish Medical Products Agency.

### Eligibility criteria
Please see box 1 for inclusion and exclusion criteria.

### Recruitment, randomisation and allocation
The study will be advertised in media and an initial screening regarding compulsive sexual behaviour, and inclusion and exclusion criteria (box 1) will be conducted when potential candidates call the national helpline PrevenTell for individuals with self-identified compulsive sexual behaviour and/or paraphilia. Persons likely to meet criteria for CSBD will receive information about the study and be invited to log into a secure web-based platform to leave a preliminary informed consent and fill in questionnaires.

Participants will thereafter attend a baseline visit at the clinic. After obtaining written informed consent (online supplemental file 1), urine samples will be collected and screened for recreational drug use (eg, amphetamine, benzodiazepines, cannabis, cocaine and opioid) and blood samples will be collected (eg, monitoring of testosterone, luteinizing hormone, glucose, electrolytes and liver enzymes) (table 1). Specific research samples will also be collected (eg, DNA extraction for genome-wide methylation analysis (EPIC) and second-generation DNA sequencing).

A psychiatrist will obtain a medical and psychiatric history, perform a physical examination including blood pressure and heart and pulmonary auscultation, as well as perform interviews with the Mini International Neuropsychiatric Interview (MINI)[18] and Columbia Suicide Severity Rating Scale (C-SSRS).[19] The study psychologist will focus on sexual behaviours by conducting a structured interview addressing the ICD-11 criteria for CSBD and the Diagnostic and Statistical Manual of Mental Disorders, Fifth Edition (DSM-5) criteria for hypersexual disorder as originally proposed for inclusion[20] and paraphilia(s). The separate psychiatrist and psychologist interviews aim

## Box 1  Inclusion and exclusion criteria

### Inclusion criteria

⇒ Meet criteria for compulsive sexual behaviour disorder according to ICD-11 and fulfil criteria for hypersexual disorder as originally proposed for inclusion in DSM-5.
⇒ Between 18 and 65 years.
⇒ Understand oral and written Swedish and have internet access.
⇒ Willing to participate in all study visits including providing blood and urine samples.
⇒ Signed informed consent form.
⇒ For fertile women: the use of a safe method of contraception during the entire study protocol.

### Exclusion criteria

⇒ Signs of hepatitis, elevated liver enzymes (>3 times over reference) or a history of liver failure.
⇒ eGFR <60 mL/min, signs or history of acute kidney failure.
⇒ fp-glucose ≥7.0 mmol/L.
⇒ Known heart disease such as angina pectoris, previous heart failure or heart attack.
⇒ Other serious physical illness including diabetes mellitus, epilepsy or known ocular hypertension.
⇒ Treatment in the past month (≥1 dose) with opioids or benzodiazepines.
⇒ Treatment in the past month with oral anticoagulants such as warfarin. Intermittent treatment (max. 15 doses per week) with NSAID (eg, ibuprofen) is tolerated.
⇒ Treatment with tamoxifen.
⇒ Self-reported use of recreational drugs in the past month or positive drug verification analysis.
⇒ Alcohol dependence or risk consumption (>14 units of alcohol per week for men, >9 for women) in the past month.
⇒ Severe psychiatric disorder requiring immediate treatment such as current psychotic disorder or severe depression.
⇒ Bipolar disorder or history of hypomania.
⇒ Ongoing treatment with naltrexone or SSRI, or previous hypersensitivity reaction to either.
⇒ Change of concurrent medication or dosage in the past 3 months regarding antidepressants, ADHD medication, mood stabilisers, antipsychotics, cortisone, testosterone or dopamine precursors. Smaller adjustments may in some cases be acceptable (assessed by study psychiatrist).
⇒ Ongoing pharmacological treatment with contraindicated substances (eg, tamoxifen and metoprolol).
⇒ Pregnancy and/or breastfeeding.
⇒ Mental condition that could negatively influence either the participant's health or the scientific aspects of the study. High risk for committing sexual offence is included.
⇒ Ongoing psychotherapeutic treatment.
⇒ Participation in other studies outside ANOVA.

ADHD, attention-deficit/hyperactivity disorder; DSM-5, diagnostic and statistical manual of mental disorders, fifth edition; eGFR, estimated glomerular filtration rate; NSAID, non-steroidal anti-inflammatory agents; SSRIs, selective serotonin reuptake inhibitors.

**Table 1**  Samples analysed during the study

| Screening | | |
|---|---|---|
| S-FSH | P-ASAT | P-Sodium |
| S-LH | P-Calcium | |
| S-SHBG | fP-Glucose | Full blood cell count |
| S-Testosterone | P-GT | B-HbA1c |
| S-TSH | P-HDL Cholesterol | |
| S-T4 | P-Potassium | U-screening for substance abuse* |
| P-Albumin | P-Cholesterol | U-Pregnancy test† |
| P-ALAT | P-Creatinine | |
| P-ALP | fP-LDL Cholesterol | Research samples‡ |
| **At 4 weeks** | | |
| P-ALAT | P-GT | Full blood cell count |
| P-ALP | P-Potassium | |
| P-ASAT | P-Creatinine | U-Pregnancy test† |
| P-Glucose | P-Sodium | |
| **At 8 weeks** | | |
| P-ALAT | P-Creatinine | U-Pregnancy test† |
| P-ALP | P-Sodium | P-Fluoxetine or U-Naltrexone§ |
| P-ASAT | Full blood cell count | |
| P-GT | B-HbA1c | Research samples‡ |
| P-Potassium | | |

*Amphetamine, benzodiazepines, buprenorphine, cannabis, cocaine, fentanyl, methadone, oxycodone, tramadol and other opioids.
†Assigned female at birth only.
‡ E.g. DNA extraction for genome-wide methylation analysis (EPIC) and second-generation DNA sequencing, oxytocin pretreatment and post-treatment.
§Depending on randomisation.

to determine eligibility criteria in an unbiased manner. If unmatched opinions, the participant will not be included.

The research nurse will open envelopes for randomisation and supply enrolled participants with either naltrexone or fluoxetine accordingly. The allocation sequence is created by an independent investigator at the Karolinska Trial Alliance, a research centre that supports clinical trials. Participants will start treatment if results on blood chemistry are adequate.

The study drugs will be provided by the Karolinska University Hospital's pharmacy.

Naltrexone (AOP Orphan Pharmaceuticals): initial dose 25 mg per day, and if tolerated will be augmented to 50 mg per day after 3–5 days.

Fluoxetine (Orion Pharma): Initial dose 20 mg per day, and if unsatisfactory symptom reduction and tolerated will be augmented to 40 mg per day after 4 weeks.

In Sweden, The Dental and Pharmaceutical Benefits Agency, a central government agency determines what pharmaceutical product shall be subsidised by the state. The agency also recommends manufacturers for specific drugs based on for example, prize and availability. The manufacturers for naltrexone and fluoxetine in this study were chosen as they were the agency's recommended product when the application was prepared for submission.

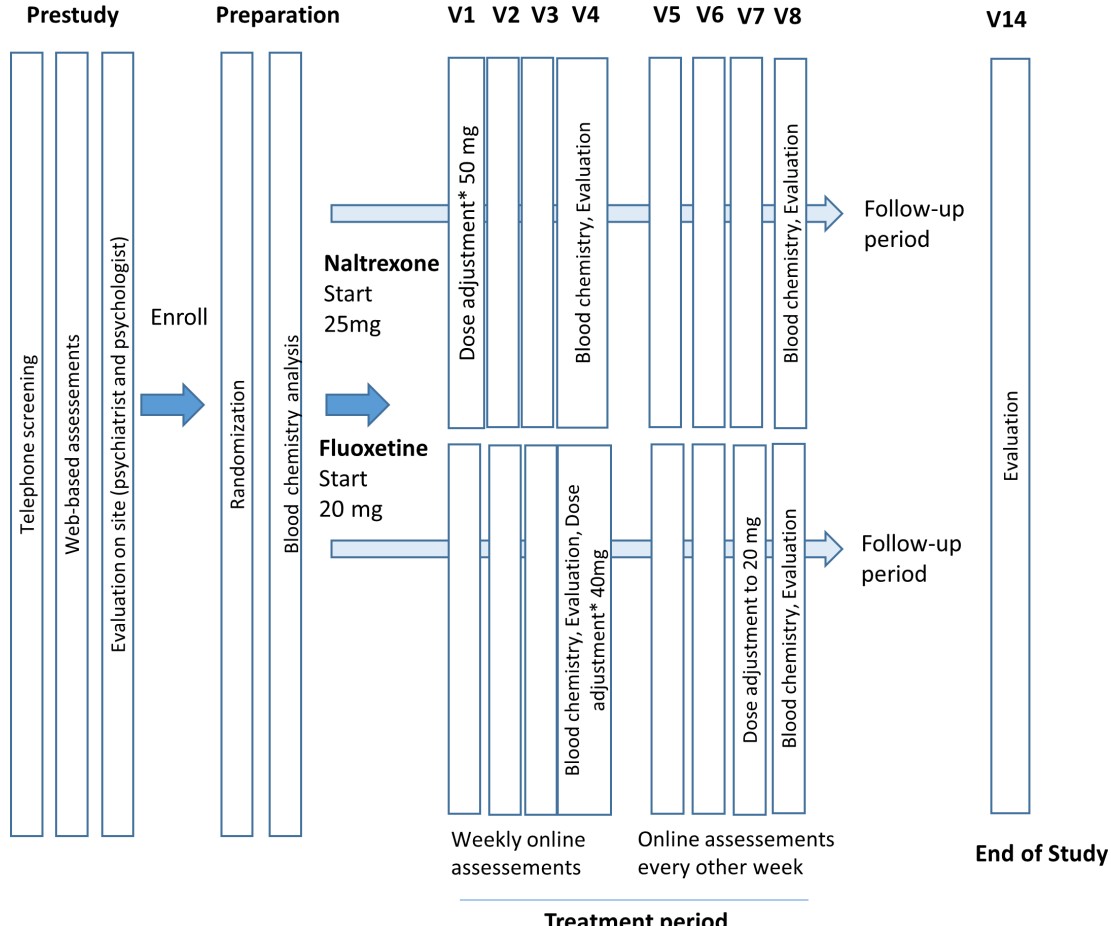

**Figure 1** Flowchart. *If unsatisfactory symptom reduction and tolerated.

## Intervention

Figure 1 illustrates study procedures. Participants will assess their symptoms weekly by filling in questionnaires online for example, if they have noticed any change in frequency or intensity of sexual urges or behaviours, how they perceive the treatment and if they experience adverse reactions. Every second week, participants will complete the main outcome measure Hypersexual Disorder: Current Assessment Scale (HD: CAS).[21] After 4 weeks, new blood samples will be collected, and participants will have a telephone consultation with the study psychiatrist to assess tolerability and psychiatric well-being. Fertile women will be required to conduct a pregnancy test before starting the study and use contraception during the study period. At end of treatment, participants will have a consultation with the study psychiatrist to assess current symptoms of CSBD, psychiatric distress and tolerability. Blood samples will be collected for chemical analysis as well as monitoring drug tablet return and the metabolites of naltrexone respective fluoxetine.

Six weeks after the end of treatment, the participants will fill in questionnaires and have a final consultation with the study psychiatrist. The visits at the clinic and the drugs will be free of charge.

## Outcome measures

The main outcome is symptom relief as assessed with HD: CAS, which measures symptom severity during the previous 2 weeks according to the suggested conceptualisation of Hypersexual Disorder to the DSM-5.[20] Corresponding scales for the ICD-11 diagnosis have not yet been developed. We chose a scale that is sensitive to changes and participants will be asked to fill it in at baseline, every second week during the treatment phase and at end of study. However, as the scale is not validated, two additional measures will be used: the Hypersexual Behaviour Inventory (HBI)[22] and the Sexual Compulsivity Scale (SCS).[23] Using unpublished data from our research centre, the correlations (Pearson's) between HD: CAS and HBI were positive at baseline and post-treatment.

We will also analyse factors (clinical, psychosocial or biological) that may be predictors of response, and whether there is any difference in participants' readiness to resume pharmacological treatment at end of study.

Furthermore, we will also evaluate tolerability using the Udvalg for Kliniske Undersogelser side effect rating scale (UKU),[24] drug accountability, adherence to treatment, and drop-out rate.

Finally, to aid in understanding CSBD, we will assess for impulsivity, suicidality and childhood adversities, and

**Table 2** Study assessments

| Objective | Assessments | Measure type | Time point |
|---|---|---|---|
| Assess changes in compulsive sexual behaviours from screening to week 8 and follow-up | Hypersexual Disorder: Current Assessment Scale (HD: CAS) | Outcome | Baseline, weeks 2, 4, 6, 8 and end of study |
| Assess compulsive sexual behaviours | The Hypersexual Disorder Screening Inventory (HDSI) | Eligibility | Baseline, and end of study |
| Assess changes in compulsive sexual behaviours from screening to week 8 and follow-up | The Hypersexual Behaviour Inventory (HBI) | Outcome | Baseline, weeks 1–4, week 6, week 8 and end of study |
| Assess changes in compulsive sexual behaviours from screening to week 8 and follow-up | Sexual Compulsivity Scale (SCS) | Outcome | Baseline, weeks 1–4, week 6, week 8 and end of study |
| Assess sexual interests | Self-assessment of sexual interests (SSI) | Eligibility | Baseline |
| Assess sexual dysfunction | International Index of Erectile Function (IIEF)* | Eligibility | Baseline |
| Assess psychiatric comorbidity | Mini-International Neuropsychiatric Interview (MINI) | Eligibility | Baseline |
| Assess changes in depression severity from screening to week 8 and follow-up | Montgomery Åsberg Depression Rating Scale—self-rating (MADRS-S) | Outcome safety | Baseline, week 8 and end of study |
| Assess change of symptoms of depression and anxiety from screening to week 8 and follow-up | Hospital Anxiety and Depression Scale (HAD) | Outcome | Baseline, week 8 and end of study |
| Assess changes in alcohol use from screening to week 8 and follow-up | Alcohol Use Disorders Identification Test (AUDIT) | Outcome | Baseline, week 8 and end of study |
| Assess use of drugs | Drug Use Disorders Identification Test (DUDIT) | Eligibility | Baseline |
| Assess pathological gambling | Gambling Disorder Identification Test (G-DIT) | Eligibility | Baseline |
| Assess impulsivity | Barratt Impulsiveness Scale (BIS-11) | Eligibility | Baseline |
| Assess symptoms of attention-deficit/ hyperactivity disorder | Adult ADHD Self-Report Scale (ASRS) | Eligibility | Baseline |
| Assess history of violence | Karolinska Interpersonal Violence Scale (KIVS) | Eligibility | Baseline |
| Assess history of childhood trauma | The Childhood Trauma Questionnaire—Short Form (CTQ-SF) | Eligibility | Baseline |
| Assess suicide ideation and suicidal behaviour throughout the study | Columbia Suicide Severity Rating Scale (C-SSRS) | Safety | Baseline, weeks 1, 4, 8 and end of study |
| Assess adverse reactions in week 8 | Udvalg for Kliniske Undersogelser side effect rating scale (UKU) | Tolerability | Week 8 |
| Question of wanting to resume pharmacological treatment, week 14 | Yes/no/do not know option | Outcome | End of study |

*Assigned men at birth only.

collect biological markers (eg, DNA and hormones) to evaluate their potential association with CSBD. A summary of assessments and biological markers is presented in tables 1 and 2, and supplemental table (online supplemental file 2). Furthermore, an extension of the study is planned to include neuroimaging data collection.

## Adherence and concomitant care
Participants can leave the study at any time without giving a reason, however any material already obtained will be analysed. Participants who miss ≥3 doses in a row will be

considered as having ceased treatment. Other reasons for discontinuation of treatment include severe adverse reactions, severe psychiatric condition where the safety of the participant or others cannot be guaranteed, initiation of treatment with non-compatible drugs or use of recreational drugs, pregnancy or start of psychotherapy.

## Sample size and statistical methods
Since there is no gold standard for the measurement of pharmacological treatment outcomes in CSBD, the sample size calculation is based on HD: CAS from a

study of Cognitive Behavioural Therapy from our clinic[25] (using expanded data available postpublication, n=76). The difference in HD: CAS prepost treatment was 4.065 with a SD of 4.74. A statistical power of 80%, an error probability of 0.05 and a difference between the groups of 3.3 (meaning an effect size of 0.7) would render a sample size of 34 participants in each group. To adjust for potential dropouts, we aim to include 80 participants. The results will be analysed using an appropriate analysis of variance model or a mixed model, depending on the distribution of HD: CAS and the extent of missing data. Participants who discontinue treatment will be included in an intention-to-treat analysis. Per-protocol analysis will also be used. To increase the study's credibility, the extent and nature of missing data will be reported for each treatment group, as well as any predictors of missing data (eg, young age).

### Monitoring

As required, an independent trial investigator from the Karolinska Trial Alliance will regularly control and quality assure the study procedures. There will be no interim analysis.

### Patient and public involvement

Patients and the public will not be involved in the design of the study other than the input from the participants and outcomes from the pilot study.[17]

### Ethics and dissemination

The study procedures will be carried out in accordance with the Declaration of Helsinki and European Good Clinical Practice Guidelines. The Swedish Ethical Review Authority and the Swedish Medical Products Agency have approved the study (ref. no. 2020-02069 and ref. no. 5.1-2020-48282). The registration of the trial is accessible at https://www.clinicaltrialsregister.eu. Study enrollment started in October 2020.

Eligible participants will be aged≥18 years and understand both oral and written Swedish. Paraphilic interests are not an exclusion criterion, whereas severe risk for sexual violence (as reported in the interviews and questionnaires, eg, participants who report that they actively interact with children for sexual purposes) is.

The study psychiatrists and psychologists are experienced in evaluating and treating patients with CSBD and paraphilic disorders, and the investigators will together decide whether the individual may be a potential study candidate.

The risks for the participants including safety aspects of the drugs have been carefully weighed against the potential benefits of conducting the study, and we believe the latter outweighs the former. The participants' health and safety will be assessed and prioritised, which may entail discontinuation of treatment. All undesirable medical events that come to our attention will be graded (mild if the symptoms are transient, moderate if temporary medical treatment for relief is needed and

severe if hospital care or prolonged medical treatment is needed) and followed until the event is resolved. Serious adverse events and suspected unexpected serious adverse reactions are handled and reported as required by the Swedish Medical Products Agency. The same insurance regulation as in ordinary medical care apply.

Research data from the web-based platform will be stored securely in servers with restricted access. Archived paper material will be stored in a locked cabinet, in accordance with current legislation and regulations. Research samples and data will be destroyed 10 years after declaring end of trial to the Swedish Medical Products Agency.

Authorship eligibility will be assured in accordance with The International Committee of Medical Journal Editors (ICMJE) guidelines. The results of the study will be presented at scientific conferences and in peer-review articles. Once the trial has been published, participants will be informed about the study results on a group level and results suitable for a non-specialist audience will be accessible on our website.

## DISCUSSION

As presented in the introduction, there is currently weak evidence for the pharmacological treatment of CSBD (for review, see Grubbs et al[26]) and hence a clear need for randomised controlled trials. However, some aspects of this study design and concept need to be discussed.

### Feasibility of enrollment

Participants might be unwilling to participate in a drug trial, hence it may be difficult to complete recruitment within the designated period of 3 years, leading to a sample too small for rending power in the analyses. In addition, the COVID-19 pandemic might complicate participants' travel to our clinic or even our capacity to conduct the study due to restrictions in the provision of healthcare (eg, limited to emergency care only). An over-representation of participants living in the Stockholm area is to be expected.

### Eligibility

We try to minimise the risk for recruiting a homogeneous group of persons with CSBD by inviting individuals with a broad range of severity of symptoms, and the use of randomisation with allocation concealment to minimise the risk for selection bias between the groups. Nevertheless, as with most studies, there is a risk for an over-representation of those with a better prognosis.

The exclusion criteria are more stringent than the inclusion criteria, partly due to the safety profiles of the drugs—a situation that regardless of study protocol would influence their use in a regular clinical setting. However, we also exclude participants with ongoing recreational drug use to minimise the risk of confounders. We also exclude participants currently being treated with interacting antidepressants and

those with a severe psychiatric disorder requiring immediate treatment, as these patients are too sick to follow our study protocol. Most patients seeking treatment at ANOVA for CSBD do not fall into these categories (particularly not the latter), but this exclusion criterion will undoubtedly lead to the inclusion of healthier individuals in the study. Although we aim to have a representative sample, a certain impact on the external validity is to be expected.

## Outcome ascertainment

As noted, our main outcome variable HD: CAS is not validated. However, it has a predefined timespan of symptom relief and has been used in previous studies at our research centre. When comparing results of HD: CAS with the validated HBI, the latter reaches a floor-effect post-treatment, whereas HD: CAS seems to better distinguish effects. Our overall judgement is thus that the advantages of the HD: CAS outweigh the disadvantages.

The HBI has an item addressing craving 'My sexual cravings and desires feel stronger than my self-discipline'. Nevertheless, we are aware that using this single item to assess craving may be a limitation.

Another limitation is that this study will not be blinded, and we will not be able to control for the anticipation effect in either group. Nonetheless, group comparisons using regression models with incorporated baseline data will still be of major value and results will be of importance to guide clinical praxis and for future meta-analyses. Finally, the study protocol duration of only 14 weeks is a major limitation, but the length was chosen for feasibility reasons. Even so, if adherence to naltrexone will be low, future studies could consider using extended-release injectable naltrexone. Fluoxetine will presumably reach steady state between weeks 4 and 8, and with this aspect in mind, the comparisons between the groups after reaching steady state will be the most informative as well as being deemed sufficient to provide important clinical knowledge.

**Acknowledgements** The authors wish to thank Tania McConaghy for language revision.

**Contributors** JS, KGÖ, CD and JJ conceived and designed the study. JS and JJ contributed to the statistical plan. JJ is grant holder. JS drafted the manuscript, and all authors contributed substantially to refinement of the manuscript. All authors approved the final version.

**Funding** This work was supported by the Swedish Research Council grant number 2020-01183 and regional agreements between ANOVA, Region Stockholm and Region Västerbotten (ALF Region Västerbotten grant number RV-929554). The pharmaceutical companies are not providing funding or being involved in any other way.

**Competing interests** JJ has participated in Advisory Board of Janssen concerning esketamine for MDD with current suicidal ideation. The other authors report no conflicts of interest.

**Patient and public involvement** Patients and/or the public were not involved in the design, or conduct, or reporting or dissemination plans of this research.

**Patient consent for publication** Not required.

**Provenance and peer review** Not commissioned; externally peer reviewed.

**ORCID iD**
Josephine Savard http://orcid.org/0000-0002-0140-4109

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
