## [Reviewer comments · BMJ Open]

ARTICLE DETAILS

TITLE (PROVISIONAL)	A Randomized Controlled Trial of Fluoxetine Versus Naltrexone in Compulsive Sexual Behavior Disorder: Presentation of the Study Protocol
AUTHORS	Savard, Josephine; Görts Öberg, Katarina; Dhejne, Cecilia; Jokinen, Jussi

VERSION 1 – REVIEW

REVIEWER	Kraus, Shane University of Nevada Las Vegas
REVIEW RETURNED	20-Sep-2021

GENERAL COMMENTS	A Randomized Controlled Trial of Fluoxetine Versus Naltrexone in Compulsive Sexual Behavior Disorder: Presentation of the Study Protocol - I was excited to see such a proposal for CSBD. This study will make important contributions to the field. I have a few minor comments for them to address. 1. Why are you including CSBD and/or paraphilia? Paraphilia is a rule out for CSBD; they can both co-occur but that's incredibly unlikely (less than five percent).2. You should drop the sexual compulsivity scale and add the CSBD-19 scale (Both et al., 2020). This is based on CSBD in ICD-11 criteria. It has a clinical cut-off of 50, and it would be good to use.3. You mentioned assessing craving. Craving for what specifically? Porn, sex, masturbation. In terms of pornography, craving measures already exist (Pornography Craving Questionnaire, Kraus and Rosenberg, 2014). Are you using a single item? Craving is a key construct to assess here especially since you're administering naltrexone which should reduce self-reported craving.4. What kind of breakdown do you suspect to get within the sample? Problematic pornography use and those with issues with casual sex?5. Amend Table 3 to include frequency of administration. Some measures are probably baseline and post-treatment. However, if you add a column, you can see what's weekly, 4 weeks, 8 weeks, etc. I would suspect you would want weekly data on behavior and craving. Depression/anxiety, etc. should be at least monthly plus you're administering an SSRI.
--

REVIEWER	Mattisson, Cecilia Institute of clinical sciences Lund, Department of Psychiatry
REVIEW RETURNED	20-Dec-2021

GENERAL COMMENTS	It would have been useful to have some participants given placebo. Still, the proposed may give desired information. It is a short time of medication with fluoxetine. Steady-state is reached after about four weeks of treatment with the drug, so the fluoxetine treatment period is rather short. Naltrexone reaches steady state faster and the differences in treatment efficacy may be difficult to evaluate. The study protocol rather short duration is also judged as a major limitation as well as that the main outcome variable CAS is not validated, but this is addressed with complimentary measures. Severe psychiatric disorders are listed among the exclusion criteria. The Mini International Neuropsychiatric Interview is not so thorough, but you have stated that most patient seeking treatment at ANOVA are rather healthy. The genetic tests should be specified, since it could be sensitive information for the participants.
---

VERSION 1 – AUTHOR RESPONSE

Reviewer: 1

Response #1: We do the recruitment from a national helpline Preventell and persons seeking help for persons with self-identified CSBD and/or paraphilias. We do agree that co-occurrence rates are low as you indicated. We aim to have a representative group of help-seeking individuals and for this reason include even persons with CSBD and paraphilic disorder(s). Persons fulfilling only criteria for paraphilic disorder(s) will not be included in this study.

Response #2: Thank you for this suggestion. We do agree that it would have been a very good scale to use. However, in the planning of this study in 2019, CSBD-19 was not available for our team. Therefore, we chose to continue with SCS and the Hypersexual Behavior Inventory (HBI) that are widely used and validated. Since we are already almost half-way in the recruitment process, we believe it is preferable to continue with these questionnaires.

Response #3: Thank you for the recommendation of the Pornography Craving Questionnaire that would indeed have strengthened the assessment of craving. The HBI has an item addressing craving “My sexual cravings and desires feel stronger than my self-discipline”. We are aware that using this single item may be a limitation. We have added a short discussion of assessing craving in the limitation section.

Response #4: In our previous study of naltrexone, the majority reported problems with masturbation and pornography use. We assume the participants will report similar problematic behaviors. Whether there will be a difference in response pattern depending of problematic pornography use or issues with casual sex will be very interesting to explore. When the study will be closed, depending on the number of persons reporting problematic pornography use we may have a possibility to explore eventual group differences.

Response #5: Thank you for this suggestion; we have added a column to include frequency of administration.

Reviewer: 2

Response #1: We do agree that a placebo-controlled trial would have been beneficial. Unfortunately, the cost for producing placebo tablets were out of budget for this academically financed study. With this major limitation in mind, we decided to proceed with this study. This limitation has been highlighted in the strength and limitation section of the revised manuscript.

Response #2: We do agree that the steady state of fluoxetine is after four weeks so the patients receiving 20mg fluoxetine will presumably be on steady state between weeks 4 and 8. With this aspect in mind, the comparisons between the groups after researching steady state will be the most informative.

Regarding the use of HD: CAS, we have several additional measurements. Both these limitations are described in the discussion section.

Response #3: We agree that the MINI is not so thorough. Mental disorders are assessed in several steps before inclusion; at the telephone screening when potential candidates call the national helpline PrevenTell. Thereafter, we assess mental disorders with validated questionnaires and at the baseline visit, the psychiatrist perform the MINI interview, as well as obtaining medical (including mental) history in a semi-structured interview.

Response #4: Thank you for pointing this out. The ethical review authority has approved the collection of DNA for genome-wide methylation analysis (EPIC) and second generation DNA sequencing. We also have approval to store blood for not yet decided analyses. A new ethical application will be conducted when these samples are to be analyzed. We have added the patient consent form as a supplementary file where the information is provided. We have also added a sentence in the manuscript, please see page 6.

VERSION 2 – REVIEW

REVIEWER	Kraus, Shane University of Nevada Las Vegas
REVIEW RETURNED	07-Feb-2022

GENERAL COMMENTS	This is a nice pre-registration for your ongoing RCT trial of fluoxetine vs. naltrexone. I am satisfied with the changes made to the paper. If you find adherence issues with naltrexone, you might consider adding a statement in the future directions about using extended-release injectable naltrexone. Your current dosage is low, so I suspect medication adherence will be good; but if not, it might be an important for outcome paper.
--

REVIEWER	Mattisson, Cecilia Institute of clinical sciences Lund, Department of Psychiatry
REVIEW RETURNED	09-Feb-2022

GENERAL COMMENTS	An important study of pharmacological treatment of CSBD. The manuscript is ok after revision.
---

VERSION 2 – AUTHOR RESPONSE

Reviewer: 1

Response: Thank you for this important point. We have added a sentence on this in the manuscript, please see page 14.

Reviewer: 2

Response: Thank you